# Essential and recurrent roles for hairpin RNAs in silencing *de novo* sex chromosome conflict in *Drosophila simulans*

Jeffrey Vedanayagam[1]☯, Marion Herbette[2]☯, Holly Mudgett[3], Ching-Jung Lin[1,4], Chun-Ming Lai[1], Caitlin McDonough-Goldstein[5], Stephen Dorus[5], Benjamin Loppin[2], Colin Meiklejohn[3], Raphaëlle Dubruille[2], Eric C. Lai[1,4]*

**1** Developmental Biology Program, Sloan-Kettering Institute, New York, New York, United States of America, **2** Laboratoire de Biologie et Modélisation de la Cellule, École Normale Supérieure de Lyon CNRS UMR5239, Université Claude Bernard Lyon 1, Lyon, France, **3** School of Biological Sciences, University of Nebraska, Lincoln, Nebraska, United States of America, **4** Weill Graduate School of Medical Sciences, Weill Cornell Medical College, New York, New York, United States of America, **5** Center for Reproductive Evolution, Syracuse University, Syracuse, New York, United States of America

☯ These authors contributed equally to this work.

\* laie@mskcc.org

**Data Availability Statement:** The raw data for the luciferase sensor assays in Fig 4A are provided in S1 Data, and the raw data for qPCR assays in Fig

## Abstract

Meiotic drive loci distort the normally equal segregation of alleles, which benefits their own transmission even in the face of severe fitness costs to their host organism. However, relatively little is known about the molecular identity of meiotic drivers, their strategies of action, and mechanisms that can suppress their activity. Here, we present data from the fruitfly *Drosophila simulans* that address these questions. We show that a family of de novo, protamine-derived X-linked selfish genes (the *Dox* gene family) is silenced by a pair of newly emerged hairpin RNA (hpRNA) small interfering RNA (siRNA)-class loci, *Nmy* and *Tmy*. In the w[XD1] genetic background, knockout of *nmy* derepresses *Dox* and *MDox* in testes and depletes male progeny, whereas knockout of *tmy* causes misexpression of *PDox* genes and renders males sterile. Importantly, genetic interactions between *nmy* and *tmy* mutant alleles reveal that *Tmy* also specifically maintains male progeny for normal sex ratio. We show the *Dox* loci are functionally polymorphic within *D. simulans*, such that both *nmy*-associated sex ratio bias and *tmy*-associated sterility can be rescued by wild-type X chromosomes bearing natural deletions in different *Dox* family genes. Finally, using tagged transgenes of *Dox* and *PDox2*, we provide the first experimental evidence *Dox* family genes encode proteins that are strongly derepressed in cognate hpRNA mutants. Altogether, these studies support a model in which protamine-derived drivers and hpRNA suppressors drive repeated cycles of sex chromosome conflict and resolution that shape genome evolution and the genetic control of male gametogenesis.

## Introduction

Meiosis is a specialized germline cell division that produces haploid gametes from diploid progenitor cells. During meiosis, equal segregation of alleles is critical for faithful transmission of

4B are provided in S2 Data. The primary gel images for Fig 5A and S1 Fig are provided in S1 Raw Images. The raw RNA-seq and small RNA data used for genome browser views shown in Fig 1 are deposited in the NCBI Short Read Archive (SRA) portal under the BioProject ID SRA: PRJNA477366.

**Funding:** These funding agencies supported this research: National Institutes of Health K99-GM137077 to JV; French National Research Agency (ANR-16-CE12-0006-01) to RD; National Institutes of Health R01-GM123194 to CM; National Institutes of Health (R01-GM083300, R01-HD108914, P30-CA008748) and Binational Science Foundation BSF-2015398 to ECL. The funders had no role in study design, data collection and analysis, decision to publish, or preparation of the manuscript.

**Competing interests:** The authors have declared that no competing interests exist.

**Abbreviations:** FISH, fluorescence in situ hybridization; hpRNA, hairpin RNA; IC, individualization complex; siRNA, small interfering RNA; SNBP, sperm nuclear basic protein; SR, sex ratio; SV, seminal vesicle.

genetic information and for maintenance of euploid genome integrity. While equal segregation is the norm, exceptions can occur in the form of meiotic drive. Meiotic drive results when normal meiosis is compromised by selfish genetic elements, DNA sequences that manipulate meiosis to gain a biased transmission advantage to the next generation. Over the last few decades, meiotic drive systems have been documented across diverse eukaryotes, including plants, fungi, insects, and mammals [1–3]. Thus, the status quo of Mendel's law of segregation is frequently breached.

Chromosomal sex determination provides unique evolutionary opportunities for meiotic drive elements that differ from autosomal systems. Unlike most autosomal homologs, sex chromosomes typically harbor highly distinct genetic and sequence content, which enables meiotic drive loci to distinguish the chromosome where they reside from the alternative homolog [4–6]. Across many animal species, parents maximize fitness by investing equally in male and female offspring, and consequently natural populations usually maintain an equilibrium sex ratio determined by the relative costs of producing male versus female offspring [7,8]. If this cost is equal, the equilibrium sex ratio will be 1:1. However, if populations deviate from this equilibrium, selection then favors alleles which produce offspring of the rarer sex, ultimately restoring the population to the Fisherian equilibrium [9]. These considerations suggest that meiotic drive elements located on sex chromosomes may be countered by unlinked loci across the genome. Mutations on the targeted homolog that escape or suppress the driver, as well as at autosomal loci that restore the Fisherian sex ratio and/or recover male fertility lost from gamete destruction can propagate cycles of coevolution between drivers and suppressors.

By biasing transmission of the X or Y, meiotic drive loci located on sex chromosomes generate skewed progeny *sex ratio* (SR). This makes sex-chromosome meiotic drive much easier to detect than autosomal drive, especially in biological systems without genetic or genomic resources [10]. However, the molecular identities of many endogenous drive factors and their modus operandi of mediating sex chromosome bias remain mostly mysterious. The insect clade has been particularly effective for revealing SR drive systems, including in numerous *Drosophila* species, stalk-eyed flies, butterflies, and mosquitoes [6,10–13]. Among these, analysis of *Drosophila simulans* (*Dsim*) suggested 3 distinct SR drive systems in this individual species [5,14], while none have yet been discovered in the major experimental model *D. melanogaster* (*Dmel*), a very close relative of *Dsim*.

Two of these SR systems were discovered through experimental introgression between *Dsim* and 2 other closely related *Drosophila*. In the Winters system, introgression of an autosomal *Drosophila sechellia* (*Dsech*) locus into the *Dsim* background revealed a pair of distorter loci on the X chromosome named *Distorter on X* (*Dox*) and its paralog *Mother of Dox* (*MDox*), which mediate SR drive by an as-yet unknown mechanism [15]. The autosomal suppressor locus that counteracts *Dox* to restore balanced sex ratio was mapped to a hairpin RNA (hpRNA) encoding locus named *Not much yang* (*Nmy*) [16]. In the Durham system, autosomal introgression of *Drosophila mauritiana* (*Dmau*) sequences into *Dsim* caused both loss of fertility and female-biased progeny, suggesting another autosomal SR suppressor named *Too much yin* (*Tmy*). *Tmy* was hypothesized to suppress a putative X-linked meiotic drive locus [17] that remains to be functionally identified.

We previously reported that *Nmy* encodes hpRNA-class small interfering RNAs (siRNAs) generated by Dcr-2/AGO2, which directly repress *Dox* and *MDox* (**Fig 1A**) [18]. We discovered this network of X-linked distorters and hpRNA suppressors to be more extensive than previously appreciated, by identifying a novel hpRNA in the genetic interval defined by *tmy* introgressions. The *Tmy* hpRNA contains regions of 80% to 90% sequence identity with *Nmy* and is capable of repressing *Dox* and *MDox in trans* [18], indicating a mechanistic link between the Winters and Durham SR systems. Moreover, recent advances in genome

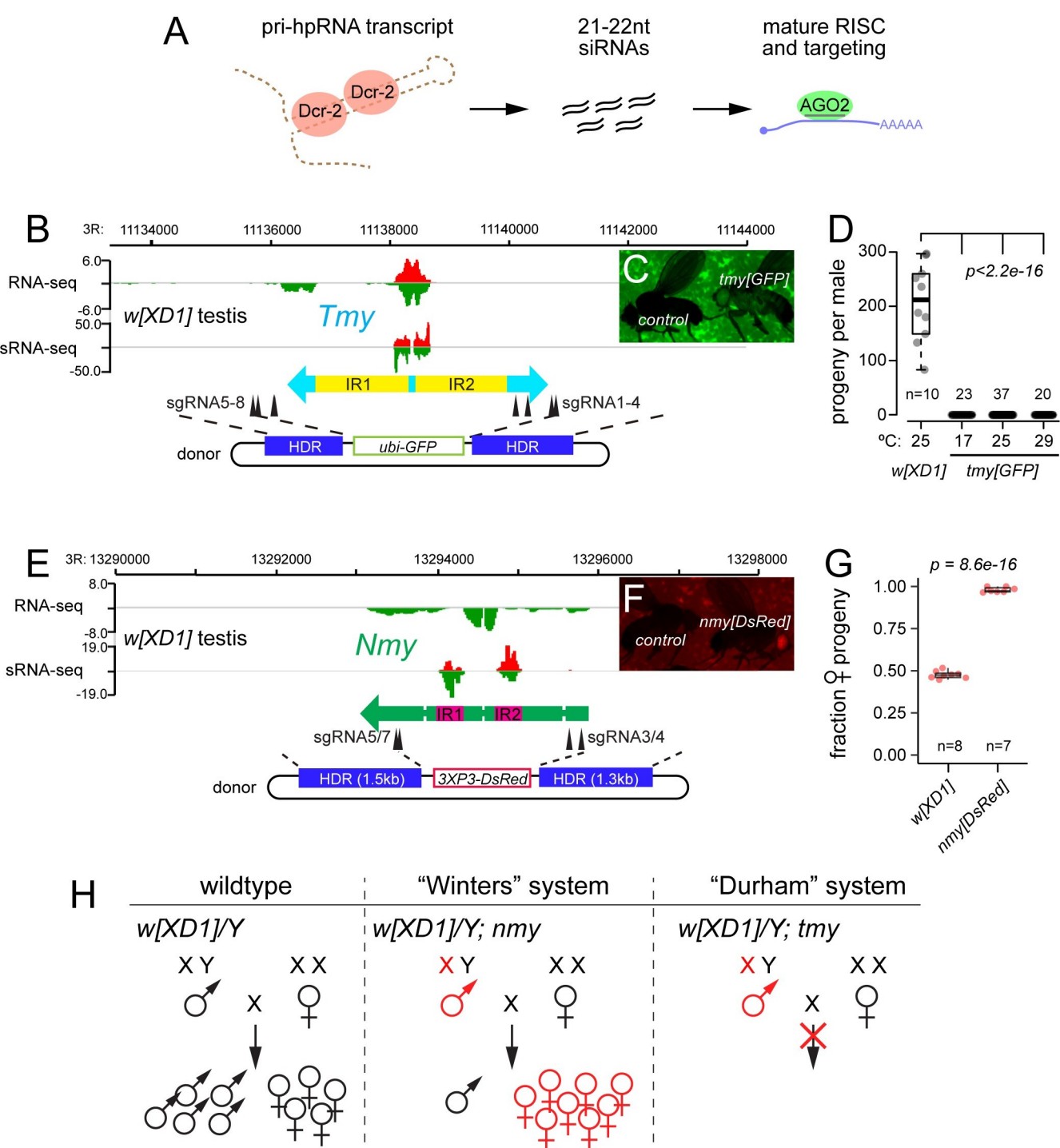

**Fig 1. Deletions of *Tmy* and *Nmy* hpRNAs reveal distinct and profound male reproductive defects.** (A) Biogenesis of hpRNAs into siRNAs via core RNAi factors Dcr-2/AGO2. We used CRISPR/Cas9 and multiple gRNAs to generate *Dsim tmy* (B-E) and *nmy* (F-J) deletion alleles for phenotypic analysis. (B) *Tmy* genomic region showing RNA-seq and small RNA-seq data from *w[XD1]* in the vicinity of the *Tmy* hpRNA. We used multiple gRNAs to delete *tmy* and replace it with a *ubi-GFP* marker, which is visible throughout the animal (C). (D) Assays of male reproduction reveal complete sterility of *tmy[GFP]* mutants at 3 different temperatures. (E) *Nmy* genomic region showing RNA-seq and small RNA-seq data from *w[XD1]* in the vicinity of the *Nmy* hpRNA. We used multiple gRNAs to delete *nmy* and replace it with a *3xP3-DsRed* marker (F). (G) Assays of male reproduction reveal nearly complete loss of male progeny by *nmy[DsRed]* mutants at 18°C. (H) Summary of *nmy* ("Winters" suppressor) and *tmy* ("Durham" suppressor) knockout phenotypes in the *w[XD1]* background. Statistical tests were Wilcoxon rank-sum test of differences between medians. hpRNA, hairpin RNA; siRNA, small interfering RNA.

assemblies [19] revealed further expansion and rapid evolutionary divergence of the X-linked Dox gene family and autosomal *Nmy*/*Tmy*-related hpRNAs in *Dsim*, *Dmau*, and *Dsech* [20,21]. Together, these findings point to ongoing and rampant sex chromosome conflicts across these 3 *simulans*-clade species.

Despite these recent advances, critical questions regarding both the mechanisms and evolutionary history of sex ratio distortion in *D. simulans* remain unanswered. First, what is the functional and evolutionary relevance of co-regulation of *Dox* and *MDox* by the *Nmy* and *Tmy* hpRNAs [18]? Does the *Tmy* hpRNA suppress an X-linked driver other than *Dox*/*MDox*, as implied by the original semi-sterile phenotype? Second, is focal deletion of the *Tmy* hpRNA sufficient to cause the *tmy* mutant phenotype? In particular, the original *tmy* introgression lines replaced megabases of *Dsim* sequences with homologous *Dmau* regions, and even the minimal *tmy* mutant region defined by partially overlapping introgression is approximately 80 kb. Moreover, it was proposed that there are complex genetic interactions in the *tmy* region that affect male fertility [17,22,23]. Thus, more precise genetics are required to clarify the extent to which loss of the *Tmy* hpRNA is responsible for the *tmy* phenotype defined by interspecies introgressions. Third, what are the functional products of *Dox* family loci, and how do they preferentially affect Y-bearing sperm? To date, there has not been direct evidence for protein products of *Dox* family loci, and the only evidence for their derepression in drive situations is at the transcript level [18].

In this study, we address these questions using new, targeted deletions of the *Nmy* and *Tmy* loci in a standardized genetic background (*D. simulans w[XD1]*). These alleles generate distinct phenotypes: *nmy*/*nmy* males sire predominantly female progeny, while *tmy*/*tmy* genotypes are completely sterile; neither mutant allele affects viability or female fertility. Nevertheless, genetic interactions between these hpRNA mutants provide support for the notion that *Tmy* is an endogenous suppressor of sex ratio distortion. We identify key targets of *Tmy* as newly recognized X-linked paralogs of *Dox*/*MDox*, referred to as *ParaDox* genes (*PDox1/2*) [20,21]. Transgenic assays demonstrate derepression of Dox and PDox2 proteins in *nmy* and *tmy* mutant testes, respectively. Finally, using naturally occurring deletion alleles of *Dox* family loci, we show that males carrying mutant alleles at both driver and suppressors rescue both fertility and sex ratio phenotypes, underscoring the role of intragenomic conflict in driving the evolution of these sequences.

Overall, we provide evidence connecting multiple enigmatic meiotic drive systems that appear to involve different members of a single X-linked protamine-derived selfish gene family and are correspondingly suppressed by autosomal hpRNA-siRNA loci. These findings testify to rapid evolutionary dynamics of meiotic drive in the male germline and reveal a critical role of endogenous RNAi to suppress intragenomic conflict.

## Results

### Precise deletion of the *Tmy* hpRNA yields complete male sterility

Over 20 years ago, Yun Tao analyzed introgressions between closely related *simulans*-clade *Drosophila* species to identify genomic intervals underlying F1 hybrid male reproductive defects observed in crosses between *Dsim* and *Dmau* [17,23,24]. In particular, introgression of *Dmau* regions on chromosome arm 3R into *Dsim* revealed the *tmy* phenotype, which manifests both as subfertility and an excess of female progeny [17]. Although the causal interval harboring *tmy* defined by overlapping introgressions was localized to <100 kb, no candidate gene was proposed until our recent discovery that an hpRNA class endo-siRNA locus resides in the *Tmy* region [18]. hpRNAs are initially transcribed as long mRNA-like primary hpRNA transcripts, which are processed by Dcr-2 into approximately 21 to 22 nt siRNAs that program

AGO2 effector complexes for gene silencing (**Fig 1A**). The discovery that the *Tmy* hpRNA is absent from the introgressed *Dmau* region [20,21] is consistent with the hypothesis that the original phenotype is due to the loss of *Tmy*. Still, there was no direct evidence that the *Tmy* hpRNA maintains fertility and/or sex ratio balance, and unfortunately, the original *tmy* introgression lines are no longer extant.

We used multiplex gRNA strategies to delete the *Tmy* hpRNA locus and replace it with a *ubi-GFP* marker (*tmy[GFP]*, **Fig 1B**) in the parent strain *Dsim w[XD1]*. We isolated multiple independent founder alleles based on fluorescence (**Fig 1C**) and genotyped both flanks to verify on-target donor replacements as well as confirm loss of *Tmy* genomic sequences (**S1 Fig**). All of these alleles proved viable and female fertile as homozygotes or as trans-heterozygotes; however, all *tmy[GFP]* mutant genotypes were completely male sterile (**Fig 1D**). Since spermatogenesis can be temperature dependent, we further tested *tmy[GFP]* mutants at 17°C, 25°C, and 29°C. These mutants remained sterile under all of these conditions (**Fig 1D**). Thus, targeted deletion shows that a de novo hpRNA is completely essential for male fertility in the *w[XD1]* background.

## Precise deletion of the *Nmy* hpRNA yields recapitulates strong sex ratio bias

Previous studies of *nmy* mutant alleles used heterogeneous genetic backgrounds [16,18,25]. Moreover, these *nmy* alleles are unmarked, rendering genetic crosses technically challenging, especially as this species lacks balancer chromosomes. We therefore deleted the *Nmy* hpRNA in *w[XD1]* and replaced it with *3xP3-DsRed* (i.e., *nmy[DsRed]*, **Fig 1E and 1F**), permitting independent visual identification of both hpRNA alleles. We isolated multiple independent founder alleles and again genotyped both flanks to verify on-target donor replacements and confirm loss of *Nmy* genomic sequences (**S1 Fig**). All of these alleles proved homozygous viable, but recapitulated previously described male-specific reproductive phenotypes, namely that they are fertile but sire highly female-based progeny (approximately 90% at 18°C, **Fig 1G**). Thus, both Winters and Durham drive systems are fully active in *w[XD1]* and can be investigated using our *nmy* and *tmy* deletion alleles in a common genetic background (**Fig 1H**).

## Loss of *Tmy* and *Nmy* hpRNAs yields distinct cytological defects in testis

We performed cytology to assess testis defects that underlie these hpRNA mutant phenotypes. The overall cellular organization and architecture of the *Drosophila* testis, and details of meiotic divisions, are schematized in **Fig 2**. During normal spermatogenesis, the mitotic and meiotic products of germline stem cell division remain connected by cytoplasmic bridges, yielding 64 spermatids that undergo coordinated elongation. Staining for histones, F-actin and DAPI reveals the orderly arrangements of actin cones within individualization complexes (ICs) of each meiotic cluster, and the progression of nuclear morphological changes from round in spermatids to needle-shaped in mature spermatozoa (**Figs 2 and 3A**). Compared to control *w[XD1]*, *nmy[DsRed]* homozygotes execute gross aspects of spermatogenesis normally, but *tmy[GFP]* homozygotes exhibit highly disorganized cysts with scattered spermatid nuclei and ICs and they fail to deposit mature sperm in the seminal vesicle (SV) (**Fig 3A and 3B**). *tmy* mutants are similar to complete loss of RNAi activity in *ago2* mutants, in that both are sterile and fail to complete spermatogenesis, but the latter are more severe and lack ICs altogether (**Fig 3A and 3B**).

More detailed examination of mature cysts during the histone-to-protamine transition revealed that histones are removed normally in *nmy* mutants, but about half of the nuclei fail to elongate and are instead irregularly shaped (**Fig 3C and 3D**). Such defects remain during

## *Drosophila* testis and spermatogenesis

**Fig 2. Overviews of the *Drosophila* testis and male meiotic divisions.** (A) Schematic organization of the *Drosophila* testis. Spermatogenesis begins at the apical tip and progresses along in a spatiotemporal manner. The apical tip contains the hub, which contains a small number of self-renewing germline stem cells (purple). By asymmetric division, the stem cells give rise to gonial cells, which will in turn undergo 4 synchronous mitotic divisions with incomplete cytokinesis, forming a cyst of 16 interconnected primary spermatocytes. After a period of growth, all spermatocytes undergo meiosis I and II in synchrony (detailed in the inset, B) to form 64 interconnected spermatids. The 64 spermatids differentiate synchronously into mature spermatozoa (only 8 are shown for clarity), during which their nuclei undergo a morphological change from round to needle-shaped. This is achieved by replacing histones (red) with transition proteins (orange), and then by SNBPs (i.e., protamines, green), yielding nuclei with a compact and elongated shape. At the same time, each cell produces a flagellum that extends almost the entire length of the testis. At the end of the histone-to-protamine transition, spermatids are individualized by actin cones (blue) that assemble around the spermatids and move coordinately from head to tail, removing excess cytoplasm. The actin cones end up in a waste bag near the apical part of the testis. Finally, mature spermatozoa coil and are released at the proximal end of the testis into the SV. (B) *Drosophila* male meiosis. Nuclei are outlined, red shading indicates chromatin. For clarity, only 1 primary spermatocyte, out of the 16 present inside a cyst, is shown. Meiosis produces 4 haploid (1N) daughter cells from a diploid primary spermatocyte (2N). After S and G2 phases, the homologous chromosomes pair up. Three territories are formed, separating the non-homologous chromosomes from each other. Two territories are formed by the 2 autosomes (chromosomes II and III) and the third by the X, Y and fourth chromosomes. The chromosomes condense in prophase I and align on the metaphase plate. Then, homologous chromosomes are separated. At the end of meiosis I, 2 daughter cells are formed, which proceed to meiosis II to separate sister chromatids and then give 2 haploid (1N) daughter cells each. SNBP, sperm nuclear basic protein; SV, seminal vesicle.

individualization, where control spermatid nuclei are needle shaped and stain uniformly with DAPI, but roughly half of *nmy* mutant nuclei are abnormally shaped and lack uniform chromatin condensation. DNA FISH experiments labeling the sex chromosomes indicate that the affected *nmy* nuclei correspond to Y-bearing sperm (**S2 Fig**), consistent with previous reports [25]. By contrast, *tmy* mutants exhibit severe defects in meiotic nuclei. In particular, histone/α-tubulin/DAPI staining revealed fragmented chromosomes and chromatin bridges in anaphase and telophase of meiosis I and meiosis II (**Figs 2E and 2F** and **S3**). Thus, while loss of *Nmy* causes largely postmeiotic defects, loss of *Tmy* affects meiotic progression.

Overall, these analyses demonstrate that deletion of the *Nmy* and *Tmy* hpRNAs result in strong, but distinct, male reproductive phenotypes. This is striking since knockout of de novo genes in *Dmel* rarely results in detectable phenotypes; although it is notable that when they do, they often impair spermatogenesis [26]. Moreover, the difference between the total sterility of our targeted *tmy* deletion mutants, and the partial fertility reported in previously described *tmy* introgression lines [17], suggested that hpRNA knockout defects might be modified by genetic background. We address and confirm this hypothesis later in this study.

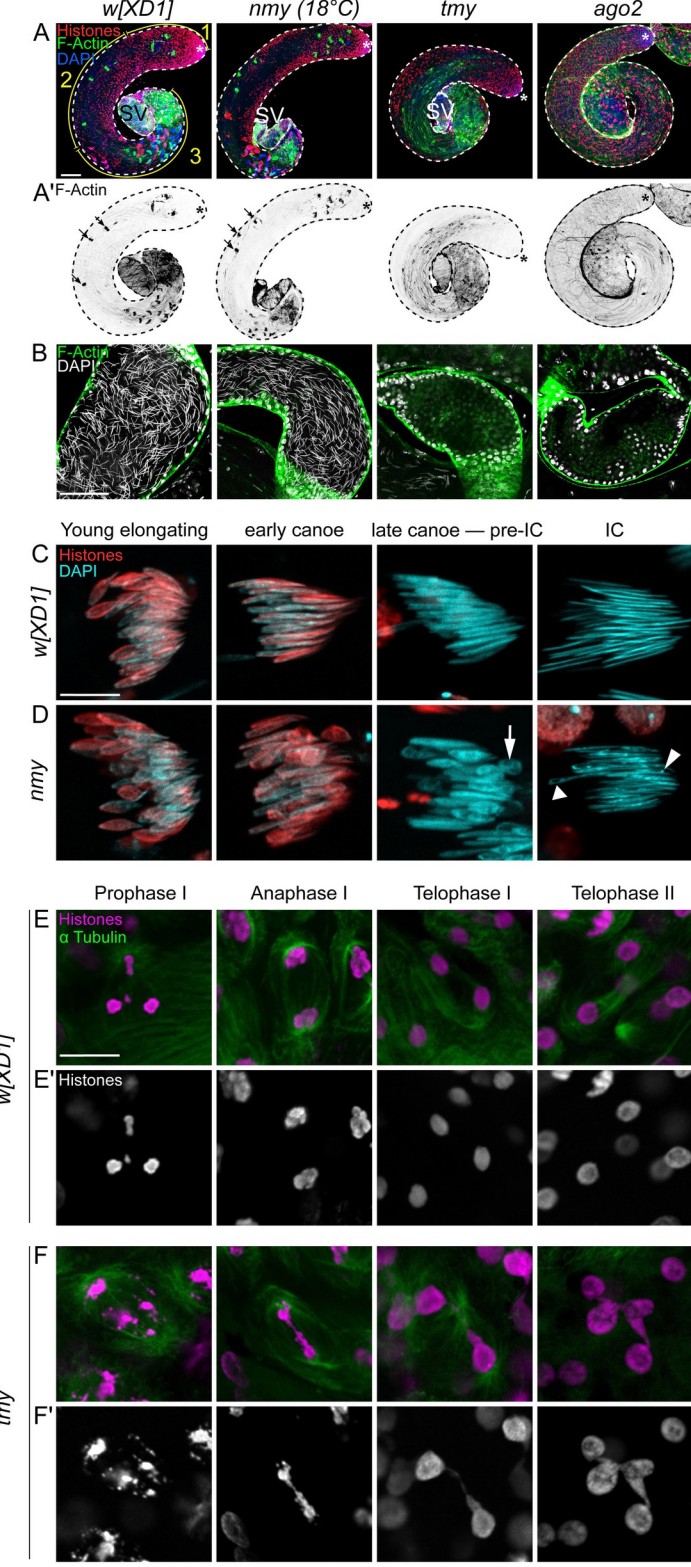

**Fig 3. Cytological basis of male reproductive defects in *nmy* and *tmy* knockouts.** (A, B) Confocal images of whole mount testes from control flies (*w[XD1]* as wild-type control) and mutants (*nmy*, *tmy*, and *ago2*) stained with a pan-Histone antibody (red) and phalloidin (F-actin green) to reveal ICs and DAPI to label nuclei (blue). (A) Whole testis images, with the stem cell regions labeled with asterisks and seminal vesicles labeled with SV. (A) Control *w[XD1]*

testis shows the orderly progression from mitotic (1), meiotic (2), and differentiation (3) regions. (B) *nmy* mutants exhibit grossly normal spermatogenesis with normal ICs (arrows, A'). Spermatogenesis is highly disturbed in *tmy* and *ago2* homozygous mutants, with disorganized cysts and aberrant (*tmy*) or absent (*ago2*) IC (A'). The empty SVs. Scale bars: 100 µm. (B) Focus on the SV shows they are filled with sperm in control and *nmy* mutants, as seen with needle-shaped DAPI staining, but are empty in *tmy* and *ago2* mutants; the large round nuclei at the borders correspond to the cells of the somatic wall. (C, D) Cysts of 64 spermatids in control (*w[XD1]*) and *nmy* mutants (grown at 18˚C). The stages of the histone-to-protamine transition indicated were determined with pan-Histone staining and nuclear shape. Histones are eliminated in both control (C) and *nmy* (D) spermatids. However, individualization of control spermatid nuclei yields regular needle shapes that stain uniformly with DAPI, whereas about half of *nmy* mutant nuclei fail to elongate and exhibit abnormal shape (arrow). In addition, DAPI staining in *nmy* mutant nuclei is uneven, with both less (triangle) or more (DAPI foci, arrowhead) dense regions, suggesting aberrant chromatin organization. Scale bars: 10 µm. (E, F) Meiosis in control and *tmy* homozygous testes stained for histones (magenta) and α-tubulin (green). The stages indicated were determined by counting nuclei number in cysts (16 for meiosis I and 32 in meiosis II). In *tmy* mutants, chromosomes are fragmented and chromatin bridges are observed in anaphase and telophase of both meiosis I and meiosis II. Scale bars: 10 µm. IC, individualization complex; SV, seminal vesicle.

## Preferred suppression of distinct Dox family targets by *Nmy* and *Tmy* hpRNAs

We reported that all *D. melanogaster* hpRNAs exhibit 1 or a few highly complementary targets [27], supporting a scenario by which hpRNAs generally derive from duplications of their target genes. We reported that *Tmy* is capable of suppressing *Dox* and *MDox* in ectopic sensor assays [18]. However, the distinct mutant phenotypes of the two *Dsim*-specific hpRNAs indicate that *Tmy* likely represses loci beyond *Dox* and *MDox*.

In a recent study, we searched the highly contiguous *D. simulans w[XD1]* genome [19] for other sequences homologous to *Tmy*. The long read assembly is particularly useful for this purpose as repetitive loci are often misassembled using short reads. Indeed, loci in this meiotic drive regulatory network are misassembled in previously available *D. simulans* genomes [18], because *Nmy* and *Tmy* encode related autosomal loci that include highly complementary inverted repeats, which also match the X-linked target genes *Dox* and *MDox*. We reported that the uncharacterized X-linked loci *GD27797A/B* are homologous to *Dox* and *MDox*, but exhibit more extensive homology to *Tmy* compared to *Nmy* [20]. Accordingly, we named them *Paralog of Dox* genes (*PDox1/2*). None of these genes are found in the *Dmel* genome, but they comprise a large family of loci that are rapidly evolving and proliferating on the X chromosomes of the 3 *simulans*-clade species [20,21].

Are *PDox* genes bona fide *Tmy* targets, and to what extent do *Nmy* and *Tmy* exhibit target specificity versus overlapping suppression? To address these questions, we first conducted heterologous gain-of-function assays to confirm these hpRNAs can directly repress *Dox* genes. We previously published that both *Nmy* and *Tmy* hpRNAs can specifically repress *Dox* and *MDox* luciferase sensors when overexpressed in a non-cognate setting (*D. melanogaster* S2 cells) [18]. We constructed *PDox1* and *PDox2* sensors and tested their response to either wild-type or mutant constructs of *Nmy* and *Tmy*. Interestingly, only wild-type *Tmy* could repress *PDox* sensors, suggesting that it might be their endogenous suppressor (**Fig 4A**).

We next isolated RNA from testis of *w[XD1]*, *nmy[DsRed]* homozygotes, and *tmy[GFP]* heterozygotes and homozygotes, and used qPCR to assess expression of *Dox*, *MDox*, and *PDox1/2* (due to their high degree of identity, we were not able to distinguish individual *PDox* loci). We observe notable target specificity in these hpRNA mutants (**Fig 4B**). Consistent with our prior tests with wild *nmy* mutant alleles [18], *Dox* and *MDox* were both derepressed in our newly generated *nmy[DsRed]* mutants; however, *PDox1/2* remained suppressed. Reciprocally, *PDox* transcripts were strongly overexpressed in homozygous *tmy[GFP]* mutants, but *Dox* and *MDox* were only subtly derepressed. None of these target genes are overexpressed in *tmy[GFP]* heterozygotes, indicating that repression by wild-type *Tmy* hpRNA is dominant. Together,

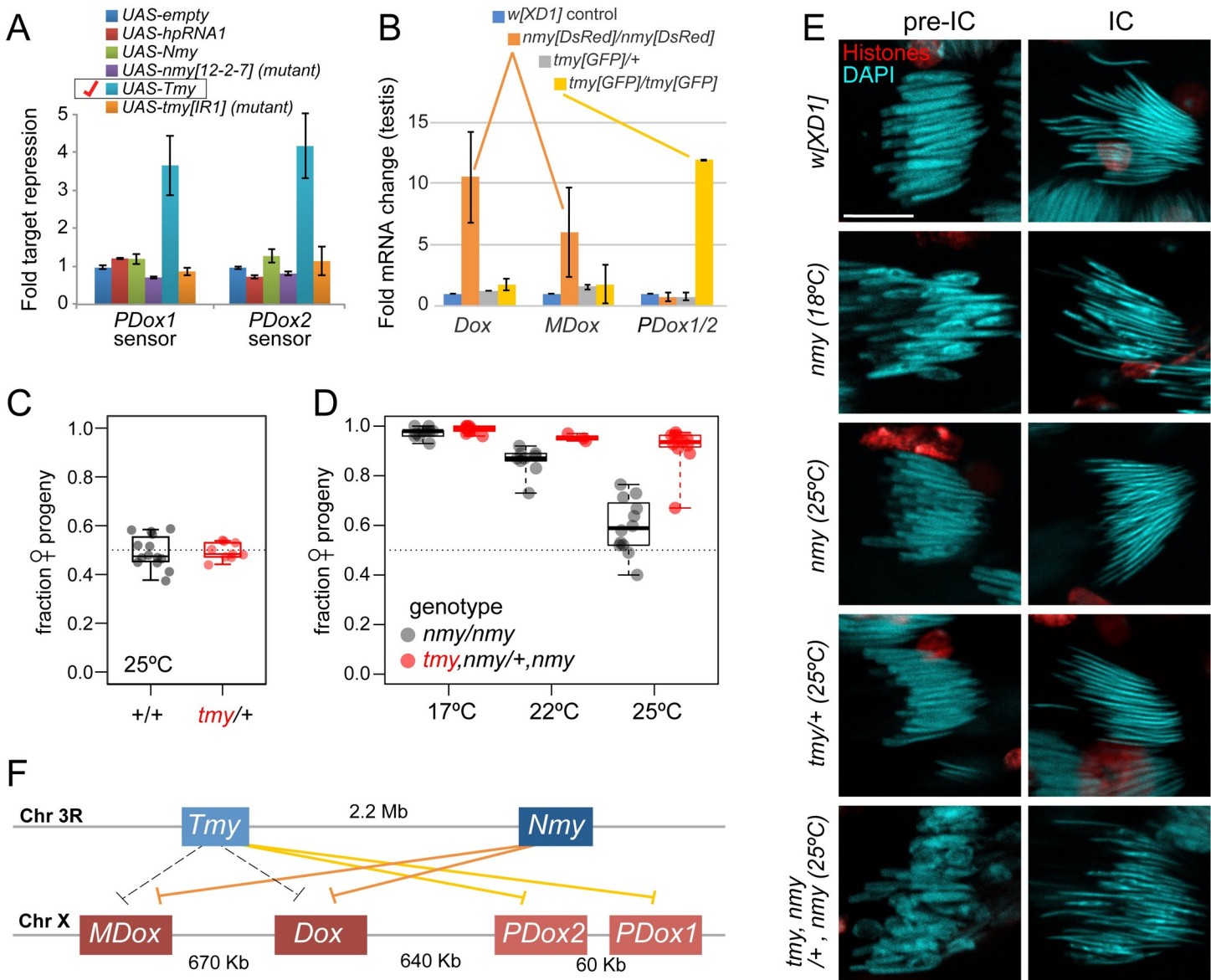

**Fig 4. Specific and overlapping roles for *Nmy* and *Tmy* hpRNAs in suppressing Dox family targets and male reproduction.** (A) Gain-of-function assays of hpRNAs and luciferase targets of *PDox1* and *PDox2* in *D. melanogaster* S2 cells. Only wild-type *Tmy* has capacity to repress both *PDox* sensors. (B) Loss-of-function assays of testis RNA from *D. simulans* hpRNA knockouts. qPCR assays show that *nmy* mutants specifically derepress Dox and MDox, while *tmy* mutants specifically misexpress *PDox* genes (amplicons could not distinguish these highly similar transcripts). (C, D) Genetic interactions support an endogenous role for Tmy in sex ratio control. (C) *tmy/+* heterozygous fathers exhibit even sex ratios in their progeny, similar to *w[XD1]* fathers. (D) *nmy* mutant males are temperature sensitive and their progeny sex ratio bias decreases from 90%–95% at 17°C to modest distortion at 25°C. Removal of 1 allele of *tmy* clearly enhances *nmy* SR defects at 22°C and 25°C. (E) Cytological analysis demonstrates that *tmy* is a dose-sensitive enhancer of *nmy* during spermatogenesis. Cysts of 64 differentiating spermatid nuclei before and during individualization were stained for histones and DAPI. In control *w[XD1]* and *tmy/+*, spermatids elongate normally. At restrictive temperature (18°C), about half of *nmy* spermatids fail to elongate normally, but this is strongly suppressed at permissive temperature (25°C). Loss of 1 *tmy* allele in *nmy* mutants at 25°C phenocopies strong *nmy* defects seen at 18°C. Scale bar: 10 μm. (F) Summary of specific and cross-regulatory repression of de novo X-linked Dox family genes by autosomal hpRNAs Nmy/Tmy. The raw data for the luciferase sensor assays are provided in S1 Data and raw data for qPCR assays in S2 Data. hpRNA, hairpin RNA.

these data support the genetic hypothesis that derepression of distinct *Dox* gene family members causes the different *nmy* and *tmy* mutant phenotypes observed here and in prior introgression studies [16,17].

## Dominant genetic interactions reveal an endogenous role of *Tmy* in sex ratio control

The complete sterility of *w[XD1]; tmy* males precluded direct assessment of *Tmy* in SR control in this genetic background. However, the Winters drive system is temperature sensitive, such that *nmy* mutant sex ratio bias is reduced at higher temperature [16]. We tested our new *nmy [DsRed]* allele and observed that progeny sex ratio bias from *nmy[DsRed]* homozygous fathers decreased as the temperature increased from 17˚C/22˚C/25˚C (**Fig 4C**). Accordingly, we sought to test if heterozygosity for *tmy[GFP]* sensitizes the loss of *Nmy* at permissive temperatures.

We first documented that *tmy* heterozygous males do not exhibit any sex ratio bias in their progeny, yielding sex ratios that are similar to control *w[XD1]* males (**Fig 4C**). Next, to test for genetic interactions between the two hpRNA mutants, we took advantage of their independent visual markers. This allowed us to generate a recombinant double mutant chromosome (*tmy [GFP], nmy[DsRed]*), which we crossed with *nmy[DsRed]/nmy[DsRed]* to generate male flies that were homozygous for *nmy[DsRed]* and heterozygous for *tmy[GFP]*. We observed that *tmy* dominantly enhanced the sex ratio bias of homozygous *nmy* mutants at both 22˚C and 25˚C (**Fig 4D**), indicating that *Tmy* is a physiological suppressor of Winters sex ratio drive. We bolstered this conclusion using cytological analysis. At 18˚C, *nmy* mutants show defects in half of the spermatids (presumably Y-bearing), but this phenotype was alleviated at 25˚C (**Fig 4E**), consistent with the temperature effects on sex ratio in this genotype (**Fig 4D**). However, *tmy [GFP], nmy[DsRed]/+, nmy[DsRed]* males showed far greater spermatid disruption than *nmy* mutants, with aberrant nuclei observed at both pre-IC and IC stages (**Fig 4E**).

Overall, these results reveal complexity in the functional roles of *Nmy* and *Tmy*. These hpRNAs appear to have distinct primary targets within the *Dox* family, with *Nmy* as the primary suppressor of *Dox/MDox* and *Tmy* as the major suppressor of *PDox1/2* (**Fig 4F**). However, we have also shown that both *Nmy* and *Tmy* are capable of repressing *Dox* and *MDox* sensors [18], while only *Tmy* is further capable of suppressing *PDox1* and *PDox2* sensors (**Fig 4A**). Taken together, these observations suggest that *Tmy* provides subsidiary repression of *Dox/MDox*, which becomes particularly overt in the *nmy* mutant context. However, further tests will be needed to distinguish whether PDox factors might also contribute to the phenotype of compound hpRNA mutants.

## Rescue of *nmy* and *tmy* knockouts by natural deletions in Dox family genes

The initial *dox* mutants [15] have been lost, precluding analysis of their genetic interactions with our new hpRNA mutants. However, deletion alleles at *Dox* and *MDox* segregate at appreciable frequencies in wild *Dsim* populations [28]. Therefore, as an alternate approach to assess the functional contribution of genetic variation at *Dox* genes to hpRNA mutant phenotypes, we inspected sequenced *Dsim* wild isolates [29,30] for variation at Dox family loci. Indeed, we identified candidate polymorphisms in several *Tmy* and *Nmy* targets. Although the repetitive nature of *Dox* loci makes their assembly challenging [18], we could validate deletions that remove both *Dox* and *MDox* in the *Dsim* strain MD15, and deletions in *Dox* and *PDox2* in *Dsim* NS40, relative to our control strain *w[XD1]* (**Fig 5A**). We confirmed these deletions by sequencing, which demonstrated that all of these natural alleles are deleted for the respective transcription units (**Fig 5B**).

If the sex ratio and/or sterility phenotypes of *nmy* and *tmy* mutants are specifically due to misexpression of *Dox* family genes, then X chromosomes lacking functional copies of *Dox* loci may restore male fertility and/or balanced progeny sex ratio when combined with hpRNA deletion alleles. Indeed *nmy[DsRed]/nmy[DsRed]* males carrying the *MD15* X chromosome,

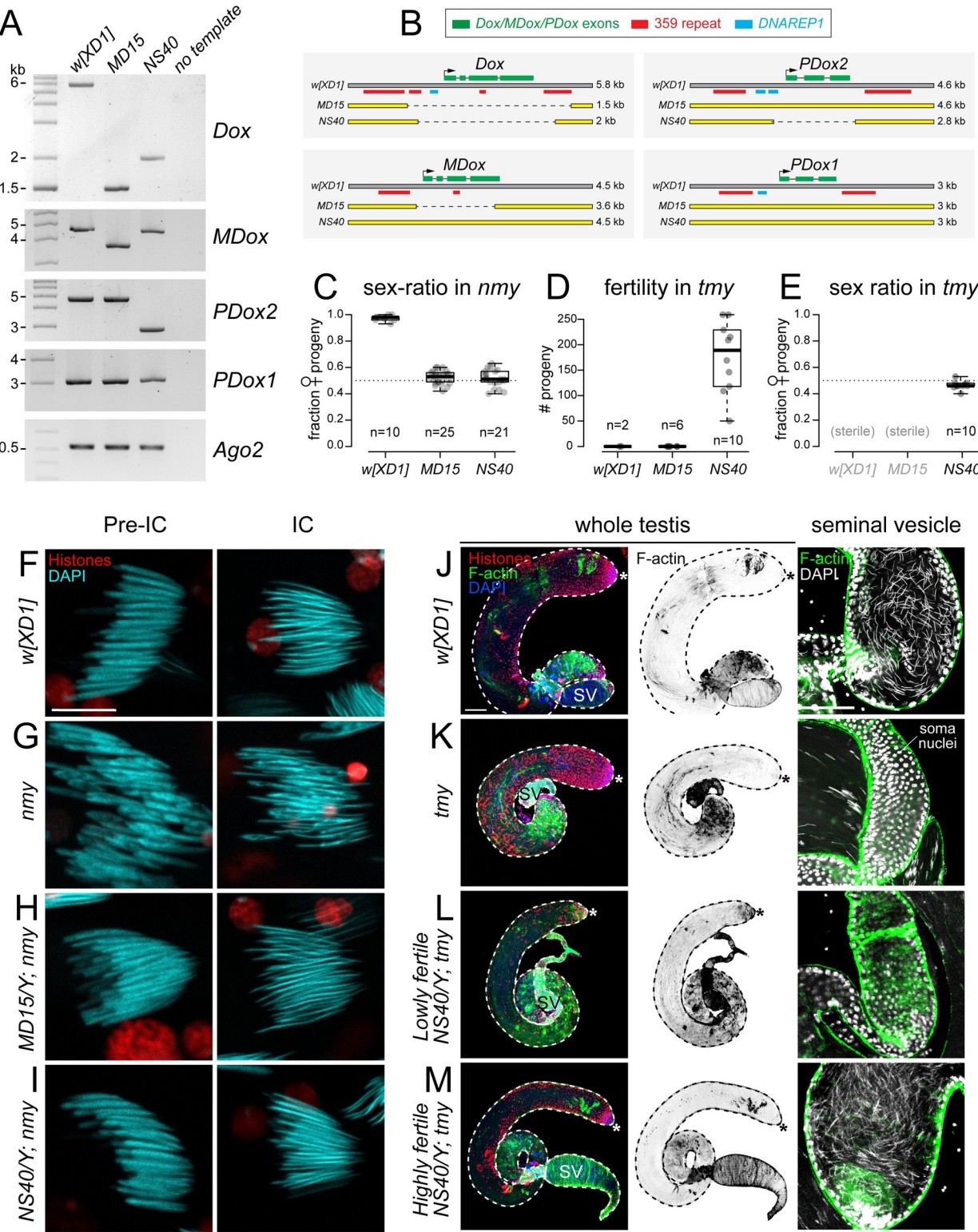

**Fig 5. Evidence that Dox family loci actively drive sex ratio and sterility phenotypes in hpRNA mutants.** (A) Screening of *D. simulans* wild lines identified X chromosomes with natural variation in Dox family loci, including deletion alleles of *Dox*, *MDox*, and *PDox2* (but not of *PDox1*). (B) Amplicon sequencing confirms that the transcribed regions of *Dox*, *MDox*, and *PDox2* are deleted in these natural alleles. (C–E) The X chromosomes of the indicated wild lines were crossed into *nmy* or *tmy* mutants and assayed for modification of these hpRNA knockout phenotypes. Each dot represents sex ratio or fertility quantification of progeny of an individual male. (C) Sex ratio assays in *nmy* knockout males

at 17°C. In control *w[XD1]*, loss of *nmy* results in near-complete loss of male progeny, but introduction of X chromosomes from either *MD15* or *NS40* restores equal sex ratio to *nmy* knockouts. (D) Fertility assays in *tmy* knockout males at 22°C. Loss of *tmy* confers complete sterility in both *w[XD1]* and *MD15*, but *NS40* restores strong fertility to *tmy* knockout males. (E) Sex ratio assays in *tmy* knockout males at 22°C. Sex ratio cannot be assessed in sterile *w[XD1]; tmy* or *MD15/Y; tmy* males, but *NS40/Y; tmy* males sire equal numbers of male and female progeny. (F–I) Effect of X suppressor chromosomes on the *nmy* loss-of-function mutants. Cysts of 64 differentiating spermatid nuclei during individualization were stained with DAPI (cyan) and an anti-pan-histone antibody (red); scale bar: 10 μm. (F) Control cysts in *w[XD1]* with normal organization. (G) At the restrictive temperature (18°C), *nmy* cysts show high frequency of misshapen nuclei. (H, I) The *MD15* (H) and *NS40* (I) X-chromosomes restore homogeneously elongated nuclei to *nmy* cysts. (J–M) Effect of X suppressor chromosomes on *tmy* loss-of-function mutants. Whole mount testes stained with phalloidin to reveal individualization actin cones (F-actin, green), an anti-pan-histone antibody (red) and DAPI (blue); scale bar: 100 μm. (J) Control *w[XD1]* shows normal testis development with mature sperm in the SV. (K) *tmy* mutant shows scattered ICs and the SV is devoid of sperm; the large round nuclei in this optical section correspond to the somatic sheath. (L) *NS40/Y; tmy* with low fertility exhibits partial restoration of germline development (*n* = 2/14). (M) *NS40/Y; tmy* with normal fertility shows corresponding rescue of normal spermatogenesis and abundant sperm in the SV (*n* = 12/14). The uncropped genotyping gels are provided in S1 Raw Images. hpRNA, hairpin RNA; IC, individualization complex; SV, seminal vesicle.

which has deletions in both *Dox* and *MDox*, produce equal proportion male and female progeny (**Fig 5C and 5D**). However, the *MD15* X chromosome, which retains intact copies of *PDox1/2*, remains sterile in *tmy* mutants, like *w[XD1]* (**Fig 5D**). Remarkably, males carrying the *NS40* X chromosome not only produced balanced progeny sex ratios in *nmy* mutants, but also were highly fertile and exhibited balanced progeny sex ratio in *tmy* mutants (**Fig 5C–5E**). As both the *MD15* and *NS40* X chromosomes bear deletions in one or both *Nmy* targets (*Dox/MDox*), but only *NS40* contains a deletion in a *Tmy* target (*PDox2*) (**Fig 5A and 5B**), these data support the model that different *Dox* genes are responsible for the distinct spermatogenesis defects in these hpRNA mutants.

We tested if rescues of male reproductive performance were associated with rescues in testis cytology. In *nmy* mutants at 18°C, we found that aberrant postmeiotic nuclear reshaping was rescued by both *MD15* and *NS40* X chromosomes (**Fig 5F–5I**). In *tmy* mutants, we separately analyzed males with low and high fertility, relative to controls (**Fig 5J–5M**). We observed cytological rescue in both classes: *tmy* mutants regained ICs in the presence of the *NS40* X chromosome regardless of fertility, although we observed some scattering of actin cones. Waste bags were more reliably detected in highly fertile males (**Fig 5L and 5M**), and mature sperm were restored to their SVs (**Fig 5L and 5M**).

Overall, we provide genetic and cytological evidence that distinct members of a rapidly evolving family of de novo protamine-derived genes drive the highly deleterious phenotypes observed in two de novo hpRNA-siRNA loci in *D. simulans*.

## Dox family loci encode protamine-like proteins with dynamic localization

While it was originally unclear whether *Dox* or *MDox* encode proteins [15], analysis of the expanded family of *Dox*-related loci revealed their homology to *Drosophila* protamine-like proteins [20,21]. This is especially evident across their HMG Box DNA binding domains, where *Dox* family proteins share more amino acids with Protamines (Mst35Ba/b) than other sperm nuclear basic proteins (SNBPs) (**Fig 6A**). However, *Dox* genes are distinct from canonical protamines. In particular, many insect SNBPs, including protamine-like proteins Mst35Ba/b and Mst77F, are enriched in cysteine residues that form intermolecular disulfide bonds [31,32]. Cross-linking of SNBPs is important for the highly compact state of sperm chromatin [33,34]. The *Dox* family proteins lack the N-terminal cysteine-rich region and their open reading frames instead are predicted to initiate just prior to the HMG box domain (**Fig 6A**).

Dox family proteins are also distinct from protamines in that they share C-terminal regions with predicted transmembrane domains (1 domain in *MDox*, *PDox1/2* and 2 in *Dox*) (**Figs 6B** and **S4**). These unique C-termini are a consequence of the complex evolutionary history of Dox loci via multiple mobilization events that fused protamine sequences with several other

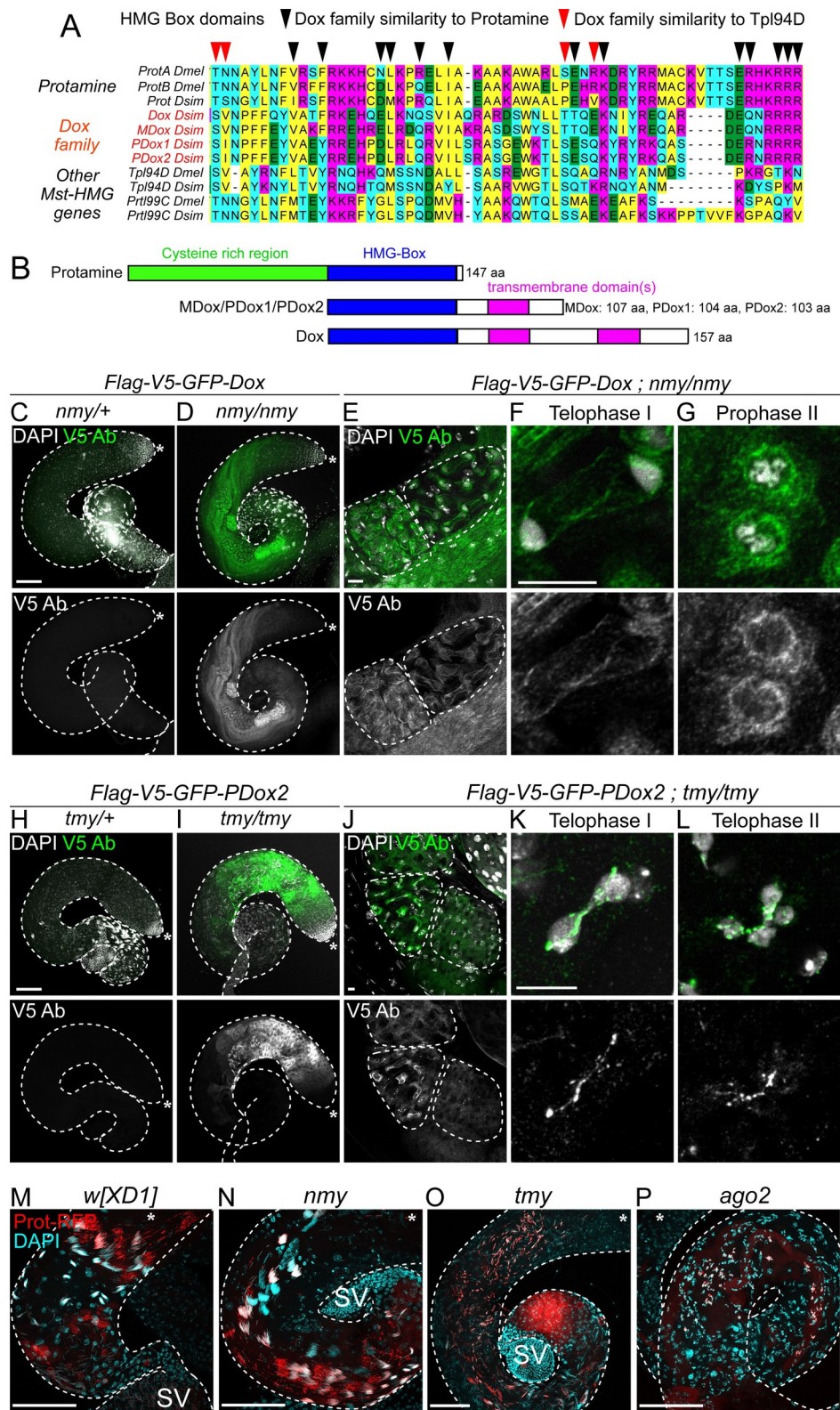

**Fig 6. Dox family loci encode protamine-like proteins that disrupt chromosome dynamics.** (A) The HMG Box domains of Dox family are more similar to protamine than other testis HMG Box proteins. (B) However, Dox loci lack the protamine N-terminus that mediates cysteine cross-linking and instead contain putative C-terminal

transmembrane domains. (C–G) Localization of Dox peptides in *nmy* mutants. Whole mount testes stained for V5 (green), and DAPI (white) shows that Flag-V5-GFP-Dox is derepressed in *nmy/nmy* compared to *nmy/+* control. In mutants, tagged Dox is initially detected in primary spermatocytes and persists through meiosis. Tagged Dox was mostly cytoplasmic where it may accumulate on cytoskeletal structures and was not detected in postmeiotic stages. (H–L) Flag-V5-GFP-PDox2 protein is derepressed in *tmy/tmy* compared to *tmy/+* control. Tagged PDox2 is detected in the cytoplasm of meiotic cells and is also enriched at cell membranes. In telophase of meiosis I and II, PDox2 forms nuclear foci and accumulates on chromatin bridges that connect meiotic nuclei. (M–P) Analysis of protamine-RFP shows normal transition in wild type (M) and *nmy* mutant (N). Spermatogenesis is defective in *tmy* but protamine is incorporated (O), while the transition to protamine is blocked in *ago2* testis (P). SV, seminal vesicle.

gene products [20,21], and they are not present in other *Drosophila* testis HMG box proteins [31]. Altogether, these observations support the model that *Dox* family genes are derived in part from protamine genes and have acquired additional novel sequences [20,21].

While these phylogenetic analyses suggest coding functions of *Dox* genes, they do not constitute definitive evidence for translated proteins. To test this, we constructed tagged genomic transgenes for *Dox* and *PDox2*, which showed no testis expression in a wild-type hpRNA genotype. However, when we tested these transgenes in hpRNA mutant genotypes, we detected *Dox* family proteins for the first time. Flag-V5-GFP-Dox (i.e., *Dox*) was derepressed in the germline of *nmy* mutants, while Flag-V5-GFP-PDox2 (i.e., *PDox2*) accumulated in the male germline of *tmy* mutants (**Fig 6C–6L**).

The subcellular localizations of Dox and PDox2 proteins were different and also distinct from the exclusively nuclear localization of tagged protamine in postmeiotic cells (**Fig 6M**). Dox protein was initially present in primary spermatocytes and persisted through meiosis (**Fig 6C–6G**). Dox was mostly cytoplasmic where it accumulated on cytoskeletal structures and was not detected in postmeiotic stages. On the other hand, PDox2 protein was initially cytoplasmic in meiotic cells and also enriched at membranes (**Fig 6H and 6I**). We observed that during telophase of meiosis I and II, PDox2 forms nuclear foci and accumulates on chromatin bridges that connect meiotic nuclei (**Fig 6J–6L**), suggesting it may cause drive via chromosome interactions.

Based on these findings, we tested if derepression of *Dox* family loci interferes with the histone-to-protamine transition. We introduced protamine-RFP transgenes into both hpRNA (*nmy* and *tmy*) and the core RNAi factor *ago2* mutant genotypes. In wild-type testes, RFP-tagged protamine is exclusively nuclear and accumulates in postmeiotic cysts and is maintained in individual mature sperm nuclei in the SV (**Fig 6M**). We find that protamine is incorporated in spermatid nuclei of both *nmy* (**S5 Fig**) and *tmy* testes (**Fig 6N and 6O**), although as noted the latter is highly disorganized and fails to generate mature sperm. Thus, by qualitative imaging, the derepression of different Dox family proteins (**Fig 6C–6L**) does not block protamine deposition. By contrast, protamines are not incorporated in the context of severely disturbed spermiogenesis in *ago2* mutant testis (**Fig 6P**), indicating that net derepression of endo-siRNA targets is associated with impaired histone-to-protamine transition.

Overall, these assays do not support the simple model that SR bias and sterility defects in hpRNA mutants can be attributed to a block in protamine incorporation. It is possible that immunostaining does not provide sufficient resolution to report fully on alterations of chromatin organization in hpRNA mutants. We also note that it remains to be determined if non-nuclear localization of protamine-derived Dox proteins (**Fig 6C–6J**), distinct from protamine itself, is functionally relevant to spermatogenesis disruption. Alternatively, this may be a legacy from the complex mobilization events during their evolutionary emergence [20,21] and might not necessarily reflect non-nuclear activities of Dox family proteins. Nevertheless, these data provide the first evidence for protein-based roles for Dox family genes and their deregulation

upon loss of cognate hpRNA suppressors, which are important steps towards deciphering the molecular mechanisms of meiotic drive.

## Discussion

### Rapidly evolving genetics of sex ratio meiotic drive and RNAi during spermatogenesis

One of the fundamental mysteries of RNA silencing concerns the endogenous biology of RNAi, the pathway used for experimental gene silencing. Although RNAi is essential for anti-viral activity, mutants of dedicated RNAi factors in several metazoans including *D. melanogaster* exhibit relatively mild defects. This is case even though flies generate a wide diversity of endo-siRNAs [35–40].

Hints into the possible endogenous roles of *Drosophila* RNAi came with the discovery that hpRNAs, which generate the most abundant individual endo-siRNAs, are most efficiently processed in the testis, repress testis-expressed targets, and are required for wild-type spermatogenesis [27]. Moreover, hpRNA-target relationships must be short-lived, as evidenced by the fact that all *D. melanogaster* hpRNAs evolved quite recently. By analogy with the role of piRNAs in controlling TEs, we propose that hpRNAs might often repress loci involved in genetic conflict, and that both hpRNA suppressors and their target selfish genes will be taxonomically restricted or even species-specific. This hypothesis is supported by the male sterility caused by loss of the RNAi pathway in *D. simulans* [18] in contrast to *D. melanogaster* where analogous genotypes are fertile and by evidence that newly emerged hpRNAs in *simulans*-clade species repress a network of de novo genes with known or presumed roles in sex chromosome conflict [20,21].

In this study, we provide direct evidence that the recently evolved *Tmy* hpRNA is required for fertility in combination with the *w[XD1]* X chromosome and can modify sex ratio distortion in *nmy* mutant genotypes. We show that *tmy* and *nmy* knockout phenotypes can be fully suppressed by wild X chromosomes that bear natural deletions in *Dox/MDox* genes and/or in *PDox* genes. Loss-of-function alleles at *Dox* segregate at appreciable frequencies in *D. simulans* [28], consistent with relaxed selection on *Dox* due to the efficient suppression of these drivers by endogenous siRNAs. Consequently, the *Dox* drive system in natural populations likely is often cryptic, and may only manifest following innovations on the part of the drivers that allow them to evade or counter suppression, followed by coevolutionary responses at Y-linked or autosomal loci (**Fig 7**). Such a scenario of relaxed selection punctuated by bouts of coevolution could account for both the functional and genetic variation observed at *Dox* loci, as well as the rapid proliferation of both the drivers and hpRNA suppressors in the *simulans*-clade species [20,21]. Indeed, the existence of the Paris system, another independent de novo sex ratio meiotic drive system *in D. simulans* [14,41], highlights the extent of rapid evolution and escalation of sex chromosome arms races in this fruit fly species.

The inference that *PDox1/2* and *Tmy* are connected to the Winters meiotic drive system (as embodied in *Nmy and Dox/MDox*) derives from their sequence similarity and shared evolutionary origins [20,21], the excess of female progeny observed in the original *tmy* introgression studies [17], and the dominant genetic enhancement of *nmy* sex ratio by *tmy* shown here (**Fig 4**). Future genetic experiments will be required to determine whether extant *PDox1* or *PDox2* alleles have meiotic drive capability; or can enhance or support drive by *Dox* and/or *MDox* alleles; or have some other, distinct role in spermatogenesis.

The network of drivers and suppressors of SR bias and sterility that we characterize in *D. simulans* exhibit functional connections, consistent with their intertwined evolutionary history, and assign specific, fast-evolving genes to bouts of meiotic drive and suppression.

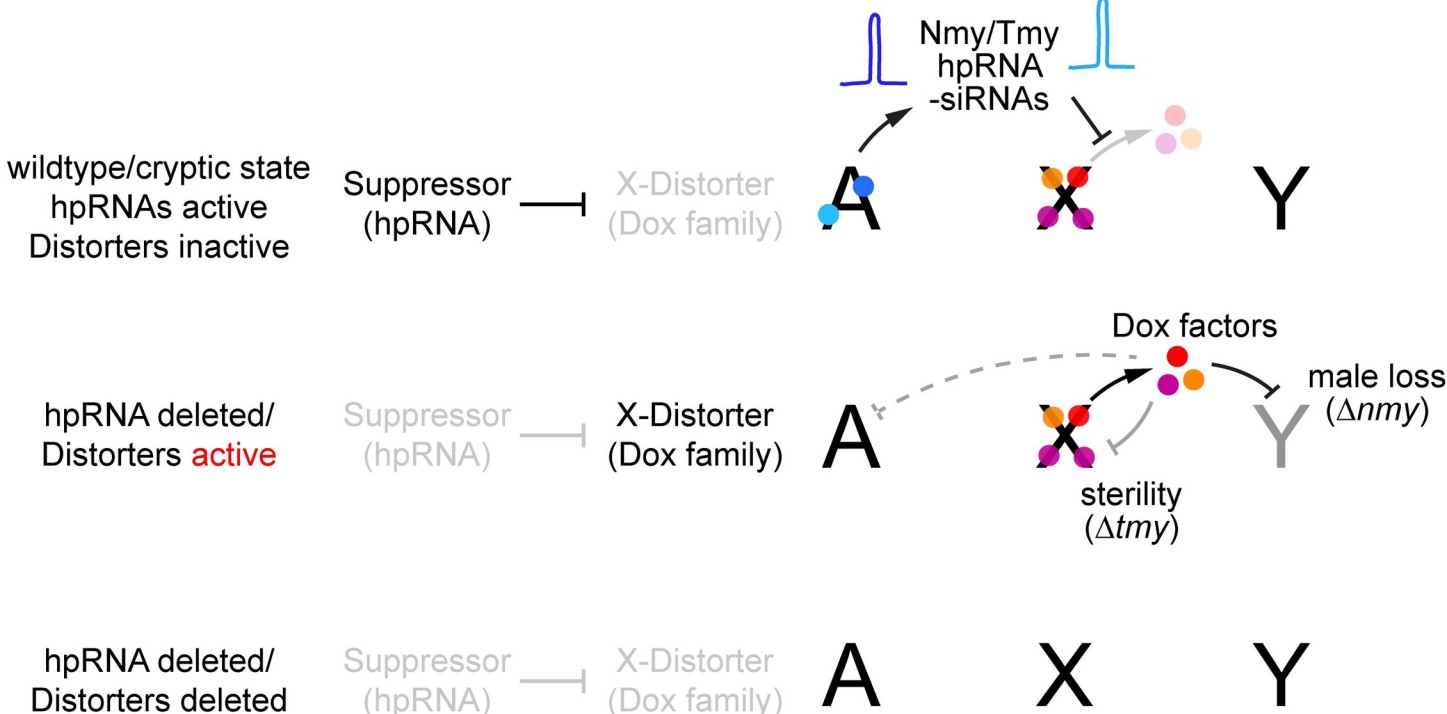

**Fig 7. Summary of intrinsic and recurrent sex chromosome conflicts and their suppression by endogenous RNAi.** (Top) Wild-type individuals harboring intragenomic conflict systems typically exist in a cryptic state, in which de novo sex-linked meiotic drive loci are silenced posttranscriptionally by autosomal hpRNA loci that generate endogenous siRNAs. Their existence of meiotic drivers is hidden in this state, since animals have normal reproductive performance, even though there can be numerous X-linked drive loci (e.g., Dox/MDox/PDox1/PDox2 in *D. simulans*). (Middle) The deleterious activities of meiotic drive loci are unleashed upon mutation of the hpRNA suppressors (e.g., Tmy and Nmy). In *D. simulans* males, loss of *nmy* results in near-complete absence of male progeny, whereas loss of *tmy* causes complete sterility. Genetic interaction tests show that Tmy protects against sex ratio drive, but evidently in the contemporary derepressed state, X-linked drivers likely off-target to the X or autosome yielding absence of female gametes as well. (Bottom) We can infer that Dox family meiotic drives are selfish, rather than contributing to normal reproduction, since removal of all of these loci restores normal fertility and sex balance to hpRNA mutants. In a sense, this returns *D. simulans* to a *D. melanogaster*-like state, which lacks the Nmy/Tmy hpRNAs as well as all X-linked Dox family genes. hpRNA, hairpin RNA; siRNA, small interfering RNA.

Extending the notion that selfish genetic elements are "motors" for evolutionary innovation [42], our data highlight that the male germline is subject to intrinsic cycles of sex chromosome conflict and resolution, involving chromatin-derived factors and small RNAs. Accordingly, we predict that additional, and perhaps many, uncloned SR meiotic drive systems will similarly involve perturbation of nuclear factors and/or suppression by hpRNAs.

### Recurrent recruitment of protamines during male meiotic drive

Although only very few loci that mediate meiotic drive have been characterized to date, it is clear that meiotic drive factors are molecularly diverse, rapidly evolving and emerge via constant innovation [1,10]. However, one emerging principle is that the histone-to-protamine transition is a critically sensitive stage of spermatogenesis, and the molecules responsible for effecting this process may frequently be co-opted to during the evolution of meiotic drive in the male germline. In addition, following on recent reports that the Dox family of meiotic drive loci originated from protamine [20,21], we provide evidence for the silencing of distinct cohorts of Dox family members in the male germline by hpRNA-siRNA loci, and that their derepression leads to both sex ratio bias and outright sterility. The subcellular localization of derepressed Dox family members is complex and includes non-nuclear accumulation, but PDox2 is clearly detected on lagging chromosomes during meiosis.

The paternal genome becomes highly condensed during sperm maturation, coinciding with the replacement of histones with protamines, in both flies [43] and mammals [44]. Indeed, the dosage and stoichiometry of protamines and related SNBPs are critical for normal spermatogenesis and/or fertility in *Drosophila* [32,33,45] and in mice [46]. Three aspects of protamine biology may make their duplication or molecular evolution particularly prone to generating meiotic drive factors. First, their expression and function are restricted to spermatogenesis, allowing them to evolve drive function with minimal pleiotropic deleterious effects on somatic and female biology. Second, they function post-meiosis, which facilitates specifically targeting one homolog while avoiding the other. Third, they interact directly with DNA; when homologous chromosomes contain different DNA content (such as the identity or extent of repetitive sequences), protamine–chromosome interactions can readily affect one homolog but not the other. This suggests how heteromorphic sex chromosomes may be particularly vulnerable to protamine-based drive; the *D. simulans* X and Y chromosomes are each approximately 40 Mb and share almost no sequence homology. However, such inter-homolog divergence can also be found on autosomes, such as the pericentromeric satellites that form the *Responder* locus of the autosomal *SD* system of *D. melanogaster* [47]. Intriguingly, although drive in *SD* is caused by a truncated RanGAP that presumably impairs nuclear import/export dynamics [47,48] but does not directly bind to DNA itself, this drive system has been recently linked to protamine dysfunction. In particular, *SD*+ spermatids do not incorporate protamine normally following histone removal [49], and knockdown of protamine enhances SD drive capacity [50]. This suggests protamine sequences have the potential to evolve alleles that might enhance or suppress *SD* drive.

*Drosophila* protamines and related SNBPs vary widely in copy number across fly species, and many are known to evolve under positive selection [20,51–53]. Indeed, it has been proposed that the rapid evolution of SNBPs is due to recurrent intragenomic conflict between sex chromosomes [53]. Our data show that de-repression of the protamine-derived *Dox* gene family can induce both SR bias and sterility and are accordingly silenced by endo-siRNA loci. As future research uncovers the molecules underlying meiotic drive systems in other species, it will be interesting to learn how frequently protamines and other SNBPs are essential components of drive, and which distinct molecules and pathways are repeatedly recruited into these selfish genetic systems.

## Materials and methods

### *Drosophila* strains

*D. simulans w[XD1]* was used as the control reference strain for analyses in the study. We used CRISPR/Cas9 mutagenesis to generate *nmy[DsRed] and tmy[GFP]* mutants. In brief, we used http://flycrispr.molbio.wisc.edu/ to design sgRNAs that delete most of the target gene, and cloned donor vectors that insert 3xP3-DsRed or ubi-GFP markers. When cloning single sgRNA into pDCC6 vector, sgRNA oligo pairs were phosphorylated by T4 PNK (NEB), annealed and ligated into BbsI digested vector using T4 DNA ligase (NEB). For cloning 4 Tmy sgRNAs into pCFD5 vector, we used the protocol from http://www.crisprflydesign.org/wpcontent/uploads/2016/07/pCFD5cloningprotocol.pdf. Homology arms of target genes were amplified from genomic DNA and cloned into pHD-DsRed or pHD-GFP-attP. All primers used to clone sgRNAs, validate sgRNA target sites, and clone donor constructs are listed in **S1 Table**.

To generate *nmy[DsRed]* or *tmy[GFP]* mutants, 100 ng/μL of pDCC6 plasmid, 100 ng/μL of pCFD5-sgRNA constructs, and 100 ng/μL donor plasmid were injected into *w[XD1]* embryos (BestGene, Chino Hills). To score for germline mutations, G0 adult flies were crossed to *w[XD1]*

virgins and F1 progeny were screened for DsRed+ eyes or GFP+ body using a Leica fluorescent microscope. Primers used to screen for on-target hybrid PCR products on the left and right flanks at target genes are listed in **S1 Table**, as are primer pairs to validate deletion of internal gene segments at each locus in homozygous mutants.

### Fertility assay

Flies were cultured on standard cornmeal molasses food. *D. simulans* bottles were set up at 25˚C to collect eggs of indicated genotypes for 2 days and transferred to designated temperatures (ranging from 17˚C to 29˚C) until adult flies emerged to obtain males for the assay. Males of each genotype were collected 1 to 2 days post-eclosion and aged 3 to 5 days, then placed singly in a vial with 3 virgin *D. simulans* w[XD1] females. After 7 days, both the male and females were discarded, and all offspring emerging from the vial were counted. All statistics and plots were made in R.

### Expression constructs

Sensor constructs. *Dox*, *MDox*, and *PDox1/2* sequences were amplified from *w[XD1]* genomic DNA into the 3′ UTR of psiCHECK plasmids (Promega). All the primers used for cloning are listed in **S1 Table**. *Dox* and *PDox2* transgenes were amplified from *w[XD1]* genomic DNA. N-terminal tagging was performed in sequential steps: upstream regions were first cloned into pCaspeR4 using NotI and BamHI sites, then coding regions and flanking downstream sequence were cloned into this vector using AscI/XhoI (*Dox*) or AscI/StuI (*PDox2*), and finally Flag-V5-GFP cassette was inserted via AscI sites.

### Luciferase sensor assay

The $1 \times 10^5$ S2 cells were seeded per well of 96-well plate and transfected with 25 ng *UAS-DsRed* or *UAS-DsRed-hpRNA* construct, 50 ng *psiCHECK* dual luciferase sensor, and 12.5 ng *ub-Gal4*. Luciferase activities were measured 3 days after transfection using Dual Glo luciferase assay (Promega) and Cytation5 luminometer (BioTek). The fold repression values were normalized to empty *psiCHECK* and non-cognate *UAS-DsRed-hpRNA* groups. Individual tests were done in quadruplicate and the averaged values from 3 biological replicate samples were subjected to statistical analysis.

### Testis cytology

For *D. simulans* analyses, 0- to 4-day-old males were dissected and the mutant testes were recognized with the empty SV and/or disrupted spermatogenesis. Whole-mount testes were stained as described [54]. Briefly, testes were dissected in PBS-T (1X PBS with 0.15% Triton), fixed in 4% formaldehyde 1X PBS for 20 min, washed 3 times in PBS-T before 4˚C overnight incubation in primary antibodies. Primary antibodies were mouse IgG2a monoclonal anti-pan-histone, clone F152 (1:1,000; Millipore), mouse IgG1 anti-alpha-tubulin clone DM1A (1:1,000; Sigma-Aldrich), mouse IgG1 V5 (1:500; Invitrogen ref#R96025), and a rabbit anti-acetyl histone H4 (1:250; Merck Millipore ref#06–598). Then, they were washed 3 times for 20 min in PBS-T prior to incubation with secondary antibodies for 2 to 3 h at room temperature. We used Alexa Fluor-coupled goat anti-mouse IGg1 and IgG2a (1:1,000; Jackson ImmunoResearch) and DyLight-coupled goat anti-rabbit (1:1,000; Jackson ImmunoResearch). Samples were then washed 3 times for 20 min in PBS-T before mounting.

For F-actin staining, testes were incubated in phalloidin diluted in 1X-PBS (1:100) and incubated at room temperature for 30 min. Testes were then washed again and incubated for

30 min at 37˚C in 2 mg/mL RNase A, rinsed and mounted in mounting medium containing 5 μg/mL propidium iodide (Sigma-Aldrich) to stain DNA. For other stainings, DNA was stained by mounting tissues directly in mounting medium (Dako) containing 10 μg/mL DAPI (Sigma-Aldrich).

### DNA fluorescence in situ hybridization (FISH)

Testes were prepared as described above, and optional immunofluorescence staining protocol was carried out first. Subsequently, fixed samples were incubated with 2 mg/ml RNase A solution at 37˚C for 10 min, then washed with PBS-T+1 mM EDTA. Samples were washed in 2xSSC-T (2xSSC containing 0.1% Tween-20) containing increasing formamide concentrations (20%, 40%, then 50%) for 15 min each. Hybridization buffer (50% formamide, 10% dextran sulfate, 2x SSC, 1 mM EDTA, 1 mM probe) was added to washed samples. Samples were denatured at 91˚C for 2 min, then incubated overnight at 37˚C. Probes used included Cy5-(AATAAAC)6 to label the Y chromosome and Cy-3-CCACATTTTGCAAATTTTGAT GACCCCCCTCCTTACAAAAAATGCG (a part of 359 bp repeats) to detect the X chromosome. Wash steps: 50% formamide+2X SSC-T for 15 min, then 20% formamide+2X SSC-T for 15 min, and finally wash with 2X SSC 3 times x 15 min. Testes were then mounted as above.

### Genotyping and qPCR analysis of *Dsim* mutants and wild lines

Because *D. simulans* lacks balancer chromosomes, and the *3xP3:DsRed* marker was not completely reliable to identify homozygous mutants, we found it necessary to genotype flies to confirm mutants prior to pooling RNA samples for further analysis. We collected 7-day-old putative *dcr-2* or *ago2* mutant males based on DsRed eye intensity and dissected their reproductive systems (testis and accessory gland) into PBS. We genotyped their respective thoraces and prepared total RNA from combined, confirmed, mutant reproductive systems using TRIzol (Invitrogen) and DNase treatment using TURBO DNA-free kit (Ambion). cDNA was synthesized using SuperScript III First-Strand Synthesis System with random primers (Invitrogen). qPCR reactions were performed with CFX96 Touch Real-Time PCR Detection System (Bio-Rad). Primers for *Dox*, *MDox*, *PDox*, and control *Rpl32* are listed in S1 Table (note that highly identical *PDox1* and *PDox2* transcripts were not confidently distinguished, so we report data as representing *PDox* genes).

To genotype natural variation across Dox family genes, we used flanking primers to amplify genomic regions of *Dox*, *MDox*, *PDox1*, and *PDox2*, and used *ago2* as a control amplicon. Genotyping primers are listed in S1 Table.

### Quantification and statistical analysis

To evaluate statistical significance for luciferase sensor assays and qPCR tests, we used unpaired Student's *t* test to calculate *p*-values. The error bars shown in fertility assays represent standard error (SEM) and standard deviation (SD) in sensor assays and qPCR tests.

### Supporting information

**S1 Fig. Genotyping validation of newly generated nmy and tmy knockouts.** Shown at left are schematics including genome browser views of the Nmy and Tmy loci and design of CRISPR/Cas9-mediated deletion of *D. simulans* hpRNA loci and replacement with fluorescent reporters. (A) *Nmy* locus with RNA-seq and small RNA (sRNA) data. Genotyping primers are indicated. (B) PCR validation of on-target knock-in of the *3xP3-DsRed* into the *nmy* locus. (C) *Tmy* locus with RNA-seq and small RNA (sRNA) data. Genotyping primers are indicated. (D)

PCR validation of on-target knock-in of *ubi-GFP* into the *tmy* locus. The uncropped genotyping gels are provided in S1 Raw Images.
(TIF)

**S2 Fig. Loss of Nmy hpRNA affects the maturation of Y-bearing sperm.** (A) Control *w [XD1]* and (B) *nmy[exf]* mutant cysts at the indicated stages of spermatogenesis, stained for DNA probes for the X and Y chromosomes (green and red, respectively) and DAPI (blue). Nuclei bearing either sex chromosome reliably elongate at the canoe stage, but those bearing the Y chromosome are defective in nmy mutants.
(TIF)

**S3 Fig. Characterization of male meiosis in tmy mutants.** Control *w[XD1]* and *tmy[GFP]* testes were stained for Histones, α-tubulin, and DAPI. The indicated panels are shown in main Fig 2, but are shown together with 1° spermatocytes and anaphase II to provide a broader view of meiotic stages. Fragmented and lagging chromosomes are observed in meiosis I and II of *tmy* mutants.
(TIF)

**S4 Fig. Relationship of Dox family factors with other testis HMG Box proteins.** (A) Amino acid alignments highlighting that Dox family proteins, similar to transition protein Tpl94D, lack N-terminal region found in insect protamines, including cysteines involved in crosslinking. However, inspection of the alignments shows that Dox family proteins are most similar to protamine in their HMG Box domains. (B) TMHMM (Transmembrane Hidden Markov Model v2) analysis shows that Dox family proteins have C-terminal putative transmembrane domains; Dox has 2 such domains. (C) The additional TM domain of Dox derives from a local duplication.
(TIF)

**S5 Fig. Protamine is incorporated into nmy mutant sperm.** Cysts from *nmy[DsRed]* heterozygous (A) and homozygous (B) testis, stained for Protamine-RFP (red) and DAPI (aqua). Transition from histone- to protamine-bearing chromatin occurs by the pre-individualization complex (IC) stage, and this occurs even in non-elongating (Y-bearing) *nmy* mutant sperm. Scale bar, 10 μm.
(TIF)

**S1 Table. Oligonucleotide sequences used in this study.**
(XLSX)

**S1 Raw images. Raw gel images used in Fig 5A and S1 Fig.**
(PDF)

**S1 Data. Raw luciferase sensor data plotted in Fig 4A.**
(XLSX)

**S2 Data. Raw qPCR data plotted in Fig 4B.**
(XLSX)

## Acknowledgments

We thank Jaeah Kim for conducting pilot studies on hpRNA genetic interactions and Himari Gunasinghe for initial genotyping tests. BL and RD acknowledge the support of Lyon SFR Biosciences (UAR3444/CNRS, US8/INSERM, ENS de Lyon, UCBL) imaging facility (PLATIM) and fly food production (Arthrotools).

## Author Contributions

**Conceptualization:** Eric C. Lai.

**Data curation:** Eric C. Lai.

**Formal analysis:** Jeffrey Vedanayagam, Marion Herbette, Holly Mudgett, Ching-Jung Lin, Caitlin McDonough-Goldstein, Stephen Dorus, Colin Meiklejohn, Raphaëlle Dubruille, Eric C. Lai.

**Funding acquisition:** Jeffrey Vedanayagam, Benjamin Loppin, Colin Meiklejohn, Raphaëlle Dubruille, Eric C. Lai.

**Investigation:** Jeffrey Vedanayagam, Marion Herbette, Holly Mudgett, Ching-Jung Lin, Chun-Ming Lai, Caitlin McDonough-Goldstein, Colin Meiklejohn, Raphaëlle Dubruille.

**Project administration:** Eric C. Lai.

**Supervision:** Stephen Dorus, Benjamin Loppin, Colin Meiklejohn, Raphaëlle Dubruille, Eric C. Lai.

**Visualization:** Jeffrey Vedanayagam, Colin Meiklejohn, Eric C. Lai.

**Writing – original draft:** Colin Meiklejohn, Eric C. Lai.

**Writing – review & editing:** Jeffrey Vedanayagam, Colin Meiklejohn, Raphaëlle Dubruille, Eric C. Lai.

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
