## [Editor Report · Decision Letter 0]

7 Nov 2022

Dear Dr Lai, 

Thank you for submitting your manuscript entitled "Essential and recurrent roles for endogenous RNAi to silence de novo sex chromosome conflict" for consideration as a Research Article by PLOS Biology.

Your manuscript has now been evaluated by the PLOS Biology editorial staff, as well as by an academic editor with relevant expertise, and I am writing to let you know that we would like to send your submission out for external peer review.

Once your full submission is complete, your paper will undergo a series of checks in preparation for peer review. After your manuscript has passed the checks it will be sent out for review. To provide the metadata for your submission, please Login to Editorial Manager (https://www.editorialmanager.com/pbiology) within two working days, i.e. by Nov 09 2022 11:59PM.

Kind regards,

Richard

Richard Hodge, PhD

Associate Editor, PLOS Biology

rhodge@plos.org

PLOS

---

## [Decision Letter · Decision Letter 1]

21 Dec 2022

Dear Dr Lai,

Thank you for your patience while your manuscript "Essential and recurrent roles for endogenous RNAi to silence de novo sex chromosome conflict" went through peer-review at PLOS Biology. Please accept my apologies for the delays that you have experienced during the peer review process. Your manuscript has now been evaluated by the PLOS Biology editors, an Academic Editor with relevant expertise, and by two independent reviewers. Please note that the Academic Editor handling your submission has also provided a review of the manuscript (labelled as 'Academic Editor Comments'). 

In light of the reviews, which you will find at the end of this email, we are pleased to offer you the opportunity to address the comments from the reviewers and Academic Editor in a revision that we anticipate should not take you very long. We will then assess your revised manuscript and your response to the reviewers' comments with our Academic Editor aiming to avoid further rounds of peer-review, although might need to consult with the reviewers, depending on the nature of the revisions.

I would also be grateful if you could address the following editorial and data-related requests that I provided below (A-F):

(A) We would like to suggest the following minor modification to the title:

“'Essential and recurrent roles for hairpin RNAs in silencing de novo sex chromosome conflict in Drosophila simulans”

(B) You may be aware of the PLOS Data Policy, which requires that all data be made available without restriction: http://journals.plos.org/plosbiology/s/data-availability. For more information, please also see this editorial: http://dx.doi.org/10.1371/journal.pbio.1001797

- Supplementary files (e.g., excel). Please ensure that all data files are uploaded as 'Supporting Information' and are invariably referred to (in the manuscript, figure legends, and the Description field when uploading your files) using the following format verbatim: S1 Data, S2 Data, etc. Multiple panels of a single or even several figures can be included as multiple sheets in one excel file that is saved using exactly the following convention: S1_Data.xlsx (using an underscore).

- Deposition in a publicly available repository. Please also provide the accession code or a reviewer link so that we may view your data before publication.

Figure 1E, 1I-J, 3A-C, 5B-D

(C) We ask that you please deposit the RNA-seq and sRNA-seq data presented in Figure 1B and 1F in a public repository, such as the GEO. Please ensure that the data is made publicly available and the accession number(s) are provided in the Data Availability Statement in the online submission form upon re-submission. 

(D) Please also ensure that each of the relevant figure legends in your manuscript include information on *WHERE THE UNDERLYING DATA CAN BE FOUND*, and ensure your supplemental data file/s has a legend.

(E) We require the original, uncropped and minimally adjusted images supporting all blot and gel results reported in the following Figures:

Figure 1D, 1H, 5A

We will require these files before a manuscript can be accepted so please prepare and upload them now. Please carefully read our guidelines for how to prepare and upload this data: https://journals.plos.org/plosbiology/s/figures#loc-blot-and-gel-reporting-requirements

(F) Please ensure that your Data Statement in the submission system accurately describes where your data can be found and is in final format, as it will be published as written there.

**IMPORTANT - SUBMITTING YOUR REVISION**

*Resubmission Checklist*

*Published Peer Review*

*PLOS Data Policy*

*Blot and Gel Data Policy*

Sincerely,

Richard

Richard Hodge, PhD

Associate Editor, PLOS Biology

rhodge@plos.org

REVIEWS:

Reviewer #1: This paper addresses the recent observation that a sex ratio drive gene (Dox) and its suppressor (Nmy) have been recurrently duplicated in the D. simulans genome, suggesting ongoing cycles of meiotic drive and suppression. The authors use reverse genetics, together with cytology and epistasis to reveal that a novel trio two Dox paralogs (ParaDox) and one suppressor (hpTmy) also impact male fertility, with ParaDox expression disrupting spermatogenesis if not controlled by Tmy. They further show for the first time that Dox family members code for proteins that are produced only in the absence of their cognate repressors, which are expressed specifically in developing sperm. This findings are very novel in suggesting for the first time how spermatotoxic proteins and repressor pairs can arise concurrently through gene duplications.

I did not review the original submission of this paper, nor did I have access to the authors response to review. I found this version of the manuscript well written, although somewhat lengthy in parts. The experiments are thorough and thoughtfully laid out. The findings are novel and I believe of interest to a general audience. I truly enjoyed reading this paper and congratulate the authors and their excellent science. 

My only significant concerns about the paper are in the discussion. First, I find there is a bit of shoe-horning when it comes to the results with ParaDox1/2. The phenotype resulting from misexpression of these two proteins is complete male sterility, which differ from that of Dox and is not necessarily indicative of meiotic drive. This observation is REALLY interesting and puzzling, since obviously ParaDox1/2 paralog would be highly deleterious and never fix if it arose on a Tmy mutant background. How do the authors envision ParaDox1/2 originally spread? Perhaps is showed drive, but a X-linked suppressor was secondarily lost after Tmy emerged? Perhaps it emerged after Tmy and persisted because it was neutral? The authors must have thought about this and I would be interested for them to better interpret this result in the discussion. As it currently stands, making a blanket claim that ParaDox1/2 and Tmy represent just another example of a SR distorter and its suppressor seems like a stretch. 

My second more minor concern is the discussion the histone to protamine transition. On pages 18-19 it is staged that "…one emerging principle is that the histone-to-protamine transition is a critically sensitive stage of spermatogenesis, and the molecules responsible for effecting this process may frequently be co-opted to during the evolution of meiotic drive in the male germline." The data in the paper show that the Dox paralogs do not impact the incorporation of protamines although they are produced in sperm at this time in the absence of repression. Thus it seems to me that while this stage may be "critically sensitive" the data in this paper do not support Dox as effectors of this transition. 

A few minor comments:

Nmy and Tmy should always be in italics since they do not code for proteins.

One page 13 I am not comfortable with the following sentence "…we do find that protamine incorporation fails completely in ago2 mutant testis (Figure 4P), indicating that net derepression of endo-siRNA targets blocks the histone-to-protamine transition."

The conclusion that can be drawn from this experiment is that ago2 is required for protamine incorporation. Whether that requirement arises from the need to repress siRNA targets is a reasonable hypothesis that would need to be tested before a conclusion could be drawn.

Reviewer #2: In this manuscript, Vedanayagam and colleagues explore in detail the biology of two hairpin RNA (hpRNA) loci that have been previously implicated as suppressors of X-linked meiotic drive elements in hybrids between two fly species D. mauritania and D. simulans. In particular, the authors investigated whether these hpRNAs are redundant in their repressive roles (shared vs unique targets) and the consequences of their loss at the cellular and molecular level in D. simulans. I think this is a very important study that helps us better understand, with unprecedented molecular detail, how genetic conflict drives genomic evolution. Overall, I found that the manuscript is well written, the logic of the experiments is clearly laid out, and the results are discussed accordingly. 

Major points

One key contribution of this study is that it allows for a fair comparison between Tmy and Nmy hpRNAs. The authors took great care to generate new mutant alleles to exclude confounding effects due to the random effects caused by introgressions, mutations, and genetic modifiers segregating in populations. 

I found the experiments described in figure 4 particularly exciting. Although not the endogenous Dox loci, the visualization of Dox and Dox-like proteins from tagged transgenes in hpRNA mutant background will definitely help characterize the biology of this mysterious protein family. Likewise, I found the rescue of hpRNA mutants using naturally occurring Dox mutant genetic variants quite elegant. I'm sorry if I missed this, but did the authors also characterize the hpRNAs loci in these wild lines (MD15 and NS40). Is it possible that in the absence of Dox and related genes, hpRNAs may have accumulated also deleterious mutations in the parental lines? Thus could help understand the evolutionary dynamics of such systems in the wild.

Regarding the Dox gene family. One criticism that I have is that the language is inconsistent and sometimes confusing. For instance, "Dox genes are not replicas of canonical protamines" (page 12), "(they) are simply protamine mimics" (page 12), "Dox family encodes illegitimate protamine copies" (figure 4 legend). To me, the term mimicry would imply convergent evolution, and I didn't see any evidence for that. I would also hesitate to call the genes "illegitimate" copies. What is legitimate for a cell after all? If the authors think these Dox families evolved by gene duplication and diversification, then they are simply "highly divergent paralogs". Related to this point, is it possible that Dox and Dox-like proteins are simply "toxins" as the ones observed in prokaryotic and eukaryotic selfish toxin-antitoxin systems? (In this case a toxin is defined as a protein that is poisonous above a certain threshold of expression but otherwise dispensable for the organism)

When tmy is mutated, there is no change in male/female ratio but males are sterile. How does this observation fit in the model that hpRNAs evolved to suppress meiotic drive of sex chromosomes? From this data, I would conclude thattmy is suppressing is not a "meiotic driver" (defined as a genetic element that biases meioses to increase its frequency in populations). Please correct me if I'm wrong, but theoretically, if I mix two populations, one "wt" and the other one carrying the dox locus (both lacking hpRNA suppressors), then the dox+ allele would be negatively selected and lost from the population.Thus, I don't think it would qualify as a meiotic driver. Thus, tmy simply seems to suppress a gene that is toxic when expressed in males during meiosis. 

Related to the previous point. Is there any reason to believe that hpRNAs suppress exclusively or mainly meiotic drive elements found in the X chromosome? As pointed out by the authors, such elements are easier to spot when located insex chromosomes because they cause deviations in the expected 1:1 sex ratio. But is there any empirical data indicating that they preferentially evolve in sex chromosomes?

Minor points

It would be good to have a bit more color consistency in Figures. For example, in Figure 1,why would panels 1E and 1I use different colors? They are both depicting the same measurement ( male fertility). Or are they supposed to match the colors assigned to Tmy (light blue) and Nmy (green)? Even though the color in 1E is not really the same as 1B. In later figures, the color code of Tmy and Nmy changes once again (see Fig 3E, for example, both are different shades of blue). I would encourage authors to make this a bit easier for reades who are not familiar with this system (it's hard to keep track of tmy/nmy and dox/Mdox/Pdox, etc)

Typo in Figure 6. "Theis existence"

The color of boxplots in Figure 3C (white versus gray) is really hard to see.

*ACADEMIC EDITOR COMMENTS*

Vedanayagam et al explore the functions of the Dox family of X-linked meiotic drivers and their suppressors in D. simulans. Tmy and Nmy were both previously implicated as autosomal suppressors of the meiotic drivers. This work provides strong experimental support that both genes are drive suppressors and provides additional insight about both genes via analyzing deletion mutants. They show that nmy homozygous mutant males are fertile and show strong drive of an X chromosome containing the Dox and MDox drive genes, but not an X chromosome without those genes. The authors find that tmy homozygous mutant males are sterile in a background that contains Dox and PDox genes, but not in a background lacking those genes. In addition, the authors find X chromosome drive in tmy/+ nmy/nmy males, but not +/+ nmy/nmy males raised at 25degrees, further supporting the role of Tmy as a drive suppressor. The authors also characterized testis morphology of both mutants and showed the localization of tagged Dox and PDox2 proteins in testes. The work makes significant progress toward understanding Dox family drivers and should guide and facilitate future work characterizing the mechanism(s) of Dox family drive. This is an interesting and important contribution to the meiotic drive and siRNA fields. The claims are mostly sufficiently supported. I point out a few concerns and suggestions for improving the work below. 

Major points: 

The abstract claims that the localization of the PDox protein indicates that it has a chromosomal function. I was not convinced the data supported this claim and suggest the abstract is reworded. My first concern is that it isn't clear to me how the authors determined that the fusion proteins were function. My second concern is that the PDox signal was not exclusively on chromosomes, as the abstract set me up to expect. Given the homology between Dox and PDox and that they both cause drive, one could argue that the cytoplasmic co-localization shared by the two proteins in meiosis is what is actually important for drive. Also, the tmy/tmy mutants seem to have a significant defect well before chromosome segregation. 

Figures 2, 4 and 5. I found much of the cytology quite data hard to see. Some of that was because of the choice of colors and presenting merged images. For example, it is hard to see red and blue on the black background. I suggest showing greyscale individual panels and then show the colors in merged images. That will also make the data accessible to color blind readers. 

In addition to the color thing mentioned above, I suggest adding cartoon explain key parts of the gonad as a new Figure 1A. I also suggest more arrows and such pointing out key features in the data. For example, I can see the seminal vesicle in WT in 1A, but I was not totally sure which little nubs were the seminal vesicle in the mutants. 

Minor Points: 

I was often confused by the genotypes assayed and suggest being more explicit when writing the genotypes in the figures and text (i.e. present homozygotes as 'nmy/nmy' or 'nmy homozygotes'). For example, on page 6, paragraph 2 says that all the trans heterozygotes were fertile then later said 'all tmy[GFP] genotypes were completely male sterile.' The latter half of the sentence sounds like tmy[GFP] heterozygotes are sterile too. 

I suggest introducing the ubi-GFP and 3xp3-DsRed markers to be more accessible to non-fly researchers. 

Figure 1G is hard to see the red on black. It would likely be easier to see if it was switched to greyscale. 

Figure 1H: I am confused by this gel. 4.5 and 3.2kb bands should not be hard to separate on a gel, but the bands are not distinguishable, and the ladder does not seem to be on the same gel. 

Is the nmy allele in supplemental figure 1 the same DsRed disruption? The label is different. 

At the bottom of page 10 it says both hpRNAs are dominant. This should be reworded to say wild-type Tmy is dominant given that data from nmy heterozygotes are not presented. 

Figure 3: I think the genotype label in 3C is incorrect. The drive observed in the tmy/+, nmy/nmy males at high temp is quite cool. To fully understand that result, I want to know if Tmy is not completely dominant at 25degrees in the background used in this study. In other words, i was not convinced the phenotype in the tmy/+, nmy/nmy was due to Dox/MDox, not PDox. Also, it is important to know if the drive phenotypes correlate with the levels of Dox/MDox/PDox expression at those temperatures. 

In Figure 3 and figure 5, the 'n' for some of the experiments is unclear (e.g. number of dots on plots is unclear). It would be good to mention those numbers in the figure legend or on the plots. 

In Figure 4, Panels E and J are not labeled or explicitly described in the legend. I was not sure what part of the gonad that was or what the white boxes are indicating. In M, the text says protamine should accumulate in the seminal vesicle but I cant see that in this image. Part of the problem is likely the colors, but the SV is also cut off in the image. 

In Figure 5, it is not clear what those PCR rxns are. I suggest a cartoon like that used in 1F to illustrate the locations of the oligos. 

For PLOS Biology's broad audience, the discussion should be broadened to include sex chromosome drivers found outside of insects.

---

## [Editor Report · Decision Letter 2]

6 Mar 2023

Dear Dr Lai,

Thank you for your patience while we considered your revised manuscript "Essential and recurrent roles for hairpin RNAs in silencing de novo sex chromosome conflict in Drosophila simulans" for publication as a Research Article at PLOS Biology. Please accept my apologies for the slight delay in getting back to you. Your revised study has been evaluated by the PLOS Biology editors and the Academic Editor who previously provided reviewer comments.

The Academic Editor has evaluated the rebuttal and the revised manuscript and notes that the revision addresses the comments from Reviewer's #1 and #2. However, he/she notes that not all of their concerns have been addressed, such as determining whether the tagged proteins are functional using genetic analyses and clarifying the reporting of the mutant phenotypes. I have provided these specific comments from the Academic Editor below my signature.

In light of these comments, we would like to invite you to revise the work to address these remaining points. Given the extent of revision needed, we cannot make a decision about publication until we have seen the revised manuscript and your response to the Academic Editor's comments. 

**IMPORTANT - SUBMITTING YOUR REVISION**

*Re-submission Checklist*

*Published Peer Review*

*PLOS Data Policy*

*Blot and Gel Data Policy*

Kind regards,

Richard

Richard Hodge, PhD

Associate Editor, PLOS Biology

rhodge@plos.org

ACADEMIC EDITOR COMMENTS

I feel the authors have addressed the comments of reviewers 1 and 2. The figures have improved and the language used is now more precise. However, I do not feel the authors have adequately addressed all of my concerns.

The outstanding major concerns:

1) The authors did not determine that the tagged proteins were functional. Given the strong phenotypes of the tmy and nmy mutants in the presence of functional Dox and Pdox2, this seems quite feasible with genetic analyses. This is also quite important if the authors want to interpret the observed localization.

Outstanding minor points:

1) The authors did not fully correct the description of the mutant phenotypes, so it was still confusing to know if they were talking about heterozygotes or homozygotes. This applies to the text and figures. Even the example text that i highlighted as being particularly confusing, was not clarified.

2) What is now Figure 4C still lacks what to me is a critical control of the tmy/+ genotype at 25 degrees to determine if Tmy is not completely dominant at that temperature. While i appreciate that full characterization of Tmy hets would require testis isolation etc, the authors could assay fertility and the fraction of female progeny with genetic crosses. If such experiments showed no fertility defects or distorted sex ratios, no further experiments would be required.

Further, the authors attribute the phenotype of the tmy/+, nmy/nmy flies to Tmy affecting Dox/MDox drive without sufficient supporting data. The data do not rule out that the phenotype is not due drive of PDox under those experimental conditions.

New minor points:

Several times, the authors refer to the ages of genes in ways i am not sure are supported by the published data. For example, the natural variants lacking Dox and Pdox2 are described as 'deletions.' Is it clear those genes are not insertions in w[XD1]? Similarly, is it clear that Dox-family-targeting hpRNAs are newly emerged in simulans and were not lost in melanogaster? Is there an out-group whose genome has been assembled well enough to address this question?

I suggest I suggest the indicated slight changes (in CAPS) to following sentence to avoid the implication that mutations are non-random: "Consequently, the Dox drive system in natural populations likely is often cryptic, and may only manifest following innovationS on the part of the drivers THAT ALLOW THEM TO evade or counter suppression, followed by co-evolutionary responses at Y- linked or autosomal loci.

---

## [Editor Report · Decision Letter 3]

21 Apr 2023

Dear Eric,

On behalf of my colleagues and the Academic Editor, Sarah Zanders, I am pleased to say that we can accept your manuscript for publication, provided you address any remaining formatting and reporting issues. These will be detailed in an email you should receive within 2-3 business days from our colleagues in the journal operations team; no action is required from you until then. Please note that we will not be able to formally accept your manuscript and schedule it for publication until you have completed any requested changes.

PRESS

Best wishes, 

Richard

Richard Hodge, PhD

Associate Editor, PLOS Biology

rhodge@plos.org

PLOS
